# Heel Height as an Etiology of Hallux Abductus Valgus Development: An electromagnetic Static and Dynamic First Metatarsophalangeal Joint Study

**DOI:** 10.3390/s19061328

**Published:** 2019-03-16

**Authors:** Rubén Sánchez-Gómez, Ricardo Becerro de Bengoa-Vallejo, Marta Elena Losa-Iglesias, César Calvo-Lobo, Carlos Romero-Morales, Eva María Martínez-Jiménez, Patricia Palomo-López, Daniel López-López

**Affiliations:** 1Faculty of Sports Sciences, Universidad Europea de Madrid, 28670 Villaviciosa de Odón, Madrid, Spain; carlos.romero@universidadeuropea.es; 2Facultad de Enfermería, Fisioterapia y Podología, Universidad Complutense de Madrid, 28040 Madrid, Spain; ribebeva@ucm.es (R.B.d.B.-V.); eva.hache2@hotmail.com (E.M.M.-J.); 3Faculty of Health Sciences, Universidad Rey Juan Carlos, 28922 Alcorcón, Spain; marta.losa@urjc.es; 4Nursing and Physical Therapy Department, Institute of Biomedicine (IBIOMED), Faculty of Health Sciences, Universidad de León, Ponferrada, 24071 León, Spain; ccall@unileon.es; 5University Center of Plasencia, Universidad de Extremadura, 10600 Plasencia, Spain; patibiom@unex.es; 6Research, Health and Podiatry Unit, Department of Health Sciences, Faculty of Nursing and Podiatry, Universidade da Coruña, 15403 Ferrol, Spain; daniellopez@udc.es

**Keywords:** hallux abductus valgus, high heel, proximal phalanx of the hallux, abduction, valgus

## Abstract

**Background:** Hallux abductus valgus (HAV) is a forefoot condition produced by extrinsic and intrinsic factors. Shoes with a high heel height and a typical narrow tip toe box can induce deviations in both the proximal phalanx of the hallux (PPH) and the first metatarsal (IMTT) bones. Nevertheless, the isolated role of heel height remains unclear in the development of HAV pathology. **Objectives:** The goal was to determine if the heel height increase of shoes without a narrow box toe could augment the PPH and IMTT deviation in frontal, sagittal, and transverse planes toward the first metatarsophalangeal joint (MPJ) and the first metatarsocuneiform joint (MCJ), respectively, during static and dynamic conditions in relation to precursor movements of HAV. **Methods:** Women with an average age of 25.10 ± 4.67 years were recruited in this cross-sectional study to assess the three planes of motion of PPH and IMTT while wearing high heels with heights at 3, 6, 9 cm and unshod conditions via sandals. The measurements used an electromagnetic goniometer device with sensors placed on medial aspects of the PPH and IMTT bones under static and dynamic conditions. **Results:** Wearing shoes with a 6 cm heel in dynamic condition may increase the PPH valgus and abduction deviation from 3.15 ± 0.10° to 3.46 ± 0.05° (*p* < 0.05) and from 1.35 ± 0.28° to 1.69 ± 0.30° (*p* < 0.001), respectively. In addition, a PPH abduction increase from 1.01 ± 0.36° to 1.31 ± 0.46° (*p* < 0.05) after wearing shoes with a 6 cm heel height was observed under static conditions. **Conclusions:** Wearing shoes with a heel height of 6 cm without a narrow box toe interference may produce PPH abduction and valgus deviations related to HAV formation.

## 1. Introduction

Hallux Abductus Valgus (HAV) is a pathological subluxation of the first metatarsophalangeal joint (IMPJ) with lateral deviation of the proximal phalanx of the hallux bone (PPH) toward abduction and valgus direction in both the transverse and frontal planes of motion, respectively. There is also deviation of the first metatarsocuneiform joint (IMCJ) through the first metatarsal bone (IMTT) toward adduction and valgus direction in both transverse and frontal planes of motion, respectively. There are occasionally bony enlargements of the first metatarsal head (also called a “bunion”) [1,2,3]. The IMPJ bears 80% of the body load without help of any structure in heel-off phase, and this makes it a more sensitive joint to biomechanics deformities such HAV [1].

The etiology of HAV can have multiple origins, and there are intrinsic and extrinsic factors in play [4,5,6,7,8,9]. Intrinsic factors include hyperpronation [5,10], soft tissue weakness [1], and hyperlaxity with medial longitudinal arch collapse [11,12]; Windlass mechanism failure [7], first ray hypermobility [8], and female sex [13,14,15] have been linked with HAV growth. As extrinsic factors, the use of high-heel shoes has also been detected as a possible cause of HAV development: The typical high-heeled shoe for women can lead to bad body repercussions and be detrimental to bone mineralization [16], rear foot instability [17], body mass center changes [18], biomechanics gait changes [19,20], and general damage health [21].

Some authors have reported increases in concentrate load under the forefoot with high heeled shoes [22,23,24,25], and this condition can predispose the subject to HAV development [26]. A few studies have speculated that the current narrow box of high-heeled shoes are not the only cause of HAV because isolated high-heel shoes might cause weight to be placed on the forefoot, and this may overstretch the toes and lead to the development of splayfoot [1,27,28]. However, this has not been reported in the literature.

The 2D [29] and 3D kinematic movements of PPH and I MTT bone deviation during gait in subjects with and without HAV has shown the relationship between the rearfoot and midfoot eversion with respect to the first ray hypermobility and the presence on HAV [10,30]; other groups have studied the kinematic effects of improvements in taping in subjects with HAV during gait [31] or the negative effects of HAV surgery to normal ambulation [32]. Other work studied the kinematics effects on IMPJ using foot orthoses that incorporated forefoot and rearfoot posting—the results showed no negative effects on mobility [33]. One prior study [34] identified kinetic evidence of wearing 5 cm-heeled shoes during gait can lead to HAV development and an increase in hallux dorsiflexion in the final push-off phase; however, no report has described the transverse or frontal plane motion of the PPH or IMTT.

Therefore, the goal of this study is to determine how the heel height affects PPH and IMTT bone deviations either in the three planes of motion in static conditions and during the dorsiflexion of the IMPJ sequence (push-off phase of the dynamic condition) regardless of toe box of the shoe. The results can show if there is some movement related to HAV deviation that is characterized by PPH in abduction (away to medial body line) in transverse plane and valgus deviation in frontal plane toward the IMPJ [2] and/or the adduction deviation of the IMTT (toward medial body line) regarding the second MTT in transverse plane and the presence of the valgus in the frontal plane toward the IMCJ [2].

## 2. Materials and Methods

### 2.1. Subjects

The institutional review board at Rey Juan Carlos University approved the study. All subjects signed the informed consent form prior to beginning this study. The following inclusion criteria were required to participate in this study [35]: (1st) to have at least 10° dorsiflexion of the ankle after complete knee fully dorsiflexion; (2nd) to have at least 30° motion in the functional subtalar joint; (3rd) to have at least 15° of motion along the mid-tarsal joint longitudinal axis; (4th) to reach at least 8 mm of non-weight bearing motion of the first ray; (5th) older than 18 years and younger than 38 years; (6th) no lower limb pathology or chronic condition at the time of recruitment and measurement; and (7th) prior experience of the use of shoes with high heel height during at least the prior year [36]. The exclusion criteria included established hallux valgus; prior history of foot and lower limb traumas, fractures, or surgeries; and congenital deformities or existing diagnoses of neurological, inflammatory, metabolic or vascular diseases. Subjects were also excluded if they showed less than 40° passive dorsiflexion of the first MPJ (as a non-weight bearing technique previously described according to Buell et al. [37]). This range has been reported to be the IMPJ dorsiflexion range used during normal propulsion [38,39] and indicates joint structural limitations. In addition, male sex was an exclusion criterion because they rarely wear high-heeled shoes.

### 2.2. Sample Size

The sample size was calculated with software from Unidad de Epidemiología Clínica y Bioestadística. Complexo Hospitalario Universitario de A Coruña. Universidade A Coruña (www.fisterra.com) to detect a significant difference in the deviation of the axes of IMTT and PPH during static and dynamic test wearing high heels versus barefoot similar in another study with 15 recruited subjects [34] where hallux dorsiflexion decreased while wearing high heels from 26.6 ± 2.33° to 22.5 ± 1.62° in barefoot condition with 80% statistical power. Here, β = 20% with a 95% interval of confidence and α = 0.05 using a 2-tailed test. Thus, at least 61 participants were required. Furthermore, assuming a loss to follow up rate of 15%, at least 71 participants were included in the study. However, recruited 80 subjects.

### 2.3. Instrumentation; Assessment of the IMTT and PPH

Three-dimensional degrees of the angular deviation of the IMTT bone movement; dorsiflexion/plantar flexion, adduction/abduction, and inversion/eversion at the first MCJ; dorsiflexion/plantar flexion, adduction/abduction, and inversion/eversion of PPH bone; and dorsiflexion/plantarflexion, adduction/abduction and inversion/eversion at the first MPJ, with and without wearing high-heeled shoes were studied during static and push-off phases using a 6Space Fastrak^®^ (Polhemus, Inc., Colchester, VT, USA) (Figure 1). This system had a 120 Hz sampling rate and a excellent validity and reliability (*r* > 0.99) [38] with ICC = 0.88–0.99 and 0.95–0.99 (SEM = 0.7–0.8 mm), respectively [40], according to previous studies; the equipment contained two sensors that send an electromagnetic field with 6° of freedom relative to the electromagnetic emitter. A 30-foot serial cable connected the sensors and emitter to a receiver module that collected, filtered, and managed all the signal’s information.

The signal from each sensor was captured by the receiver module. It was digitally transformed by the software to generate spatial orientation data from each sensor. The system showed a 8-mm static accuracy relative to the sensor position and 0.15° regarding sensor orientation; the error was 1.6% [38,41]. The relation between range vs. resolution in orientation aspect was 0.3 m to 1.16 m; in addition, the relation between range vs. orientation aspect was 0.3 to 0.0038°.

In this study, the electromagnetic emitter was 96 cm high at the midway point on a 6-m raised walkway. There were no metallic elements near the electromagnetic device or in the subject’s walking path to avoid possible interference [42]. One of the sensors was placed at the medial aspect of the head of the IMTT bone (Figure 2), and the second one was located along the medial aspect of the PPH (Figure 3) on the right foot; the Polhemus Fastrack^®^ (Polhemus, Inc., Colchester, VT, USA) did not measure angles but did measure bone displacements. Thus, we assessed the mobility of the PPH regardless of IMTT mobility to determine which areas were affected by the high heels. Both sensors were attached with tape and secured to the skin with an auto-adhesive bandage (Figure 4); this medial location was selected according to a protocol devised by Welsh et al. [43] to minimal overlying soft tissue of extensor hallucis longus tendon’s excursion [41,44] All cables were fixed via straps to the thighs with a belt.

### 2.4. Procedure

Each of 80 participants began static or dynamic study randomly after choosing a sealed envelope that assigned them to one group or another; there were 40 total in each group. They then selected four sealed envelopes with each of the four conditions of the study (unshod, 3 cm, 6 cm, and 9 cm). These dictated the order of use at random. The 40 participants of the dynamic group did the static test in randomly order with four sealed envelopes and vice versa with the 40 participants in the static test (Figure 5).

Before beginning either angular measurements, position “zero” was achieved to calibrate the assessment and determine a reference position from which to begin the angular record. We asked the participants to remain in a relaxed standing position near the generated field for a few seconds until the software determined the Reference [35].

Participants used a pair of sandals with three different prefabricated high heels (Figure 6). The sandals (Figure 7) provided a strap in the first digital web space, and the rearfoot had a semi-rigid bowl that permitted the heels to be held into the shoes. The participants used the sandals and tested the sensations with the cables before starting the study; they then walked along the walkway at a self-selected speed. When they were comfortable, they began the static or dynamic measurements depending on the order chosen by the random chance.

For the dynamic condition study, participants initiated gait for 1 m at a self-selected speed before entering the 1.5 m calibrated capture volume. They then continued walking for a further 1.5 m (two steps); this capture was repeated for 5 trials with each of 3 different types of prefabricated heel heights (3 cm, 6 cm and 9 cm) and unshod condition. Care was taken when inserting and removing the prefabricated heel height so that the sensor devices was not disturbed or displaced. For the static condition study, subjects were asked to stand on their tiptoe (to reach the maximal range of motion of the joint) for 2 s. There were five trials with each of three different types of prefabricated heel heights (3 cm, 6 cm and 9 cm) and unshod condition. The mean of the five trials was used in the posterior analysis for each static and dynamic test.

Later, a “cardan system” with three possibilities of rotation (X-Y-Z) in two ways was used to draw angular and linear information from both PPH and IMTT bones movements: mediolateral axis (X), defined dorsiflexion (DF), and plantar flexion (PF) movements on the sagittal plane as well as frontal plane inversion (VR) and eversion (VL) movements through an antero-posterior axis (Y). Movement around a craneocaudal axis (Z) was considered to be abduction (ABD) and adduction (ADD) movements on a transverse plane. The anatomical landmarks and coordinate systems were previously detailed [35,44,45]. All bone movements were recorded after considering the laboratory coordinate system. All assessments used the same heel height in contralateral extremity under the foot to maintain the body balance during these trials [46]. Independent assessment of the bone movements showed that the results were only produced by high heel effects and not by bone interference.

### 2.5. Statistical Analysis

We recorded the within-day trial-to-trial intraclass correlation coefficients (ICC) and standard errors of measurement (SEM) [47] values for the participants wearing the different heel height (T3, T6, T9) vs. unshot in each plane of motion for PPH bone and IMTT bone in static and dynamic conditions. Landis and Koch [47] proposed that coefficients below 0.20 indicate slight agreement, and coefficients from 0.20 to 0.40 indicate fair reliability. Coefficients from 0.41 to 0.60 indicate moderate reliability, coefficients from 0.61 to 0.80 indicate substantial reliability, and coefficients from 0.81 to 1.00 indicate almost perfect reliability. We considered coefficients of 0.90 or larger to reflect a sufficient magnitude of reliability because they increase the likelihood that a measure is also reasonably valid. To check the concordance correlation between high heels’ variables, Concordance Correlation Coefficient (CCC) [48] was done, where statistically significant *p*-values (<0.05) would mean a perfect correlation [48]. SEM was utilized to determine the minimal detectable change (MDC) for all evaluations. This was also considered the Reliable Change Index (RCI). We utilized the RCI as a statistical method to determine the clinical significance according to Jacobson and Truax [49].

An initial Kolmogorov-Smirnov test showed that the data was not normally distributed (*p* < 0.05). The *p*-values for multiple comparisons were corrected with a non-parametric paired Friedman test to prove that all high-heels variables were different. Bivariate correlations with a Wilcoxon test were carried out to determine whether there were significant differences between “unshod” vs. “with high heels of T3, T6 and T9” in static and dynamic conditions in FPH and IMTT bones; an alpha level of 0.01 was established for all tests of significance. All data were studied to establish the effect of heel height in 3 axes of movement with the medial line of the body taken as reference movement. In addition, Spearman’s Rho (ρ) rank correlation coefficient both in static and dynamic conditions was done to check the possible relation between the height of the heels and the different bones’ deviations. The results showed descriptive summaries as the mean ± SD. Analyses of total 96 variables led to *p*-values <0.05 (within a 95% confidence interval) that considered statistically significant. We conducted data analysis with SPSS software version 19.0 (SPSS Science, Chicago, IL, USA).

## 3. Results

None of the data were normally distributed (*p* < 0.05). The means that the 96 measured variables were different when compared between each other (*p* < 0.001). Participants in this cross-sectional study were recruited from an orthopedic clinic in Madrid (Spain) over a 2-year period (May 2015 to May 2017). Of the 163 subjects who initially volunteered to participate in the study, 68 subjects did not meet the inclusion criteria. An additional 15 subjects did not present for testing. The remaining 80 subjects participated in the study (Figure 8). The participants were only females; sociodemographic data are shown in Table 1.

All Tables are shown with corresponding values of IMTT and PPH to the static test as well as independent of the dynamic values.

The reliability of the static variables in unshod and heels of 3 cm, 6 cm and 9 cm high are summarized in Table 2. Each condition showed overall excellent reliability [50] with ICCs ranging from 0.902 to 0.997 indicating high reliability. In addition, in general terms, CCC showed a strong statistically significant correlation (*p* < 0.001) between unshod and either high heel in all movements of PPH and MTT except those related to a sagittal plane. The reliability of dynamic variables in unshod subjects for heel heights of 3 cm, 6 cm and 9 cm is summarized in Table 3. Each condition showed an overall excellent reliability [50] with ICCs ranging from 0.898 to 0.999 indicating high reliability. The CCC showed a strong correlation (*p* < 0.001) in all movements of PPH and MTT in dynamic tests, as well. The inter-rater MDC 95% values to static (Table 2) conditions ranged from 14.915° to 0.74° for sensor 1 (IMTT) and from 7.90° to 1.63° for sensor 2 (PPH); the inter-rater MDC 95% values to dynamic (Table 3) conditions ranged from 26.78° to 1.12° for sensor 1 (MTT) and from 27.16° to 2.16° for sensor 2 (PPH).

The SEM values for 6 cm high heel in static abduction of PPH (Table 2) was 0.591° and 0.782° for dynamic abduction condition (Table 3); in addition, the SEM was 2.02° for valgus movement of PPH in dynamic condition with high heels of 6 cm and 2.33° in static condition.

Static mobility grades of IMTT (sensor 1) and PPH (sensor 2) bones in unshod and 3 cm, 6 cm and 9 cm heel height are summarized in Table 4. The dynamic mobility grades of both sensor 1 and sensor 2 in unshod and 3 cm, 6 cm and 9 cm heel heights in Table 5.

Under static conditions, the PPH abduction increases from 1.01 ± 0.36° to 1.31 ± 0.46° (*p* < 0.05) after wearing shoes with a 3 cm heel. There was a statistically significant valgus of the I MTT with increasing heel heights from 1.77 ± 0.20° without high heels to 2.15 ± 0.10° with 3 cm (*p* < 0.001) high heels. There was increased varus movement from 1.50 ± 0.23 to 3.87 ± 0.20 with 3 cm heels (*p* < 0.001).

In dynamic tests, wearing shoes with 6 cm high heels led to an increase in PPH valgus and abduction deviation from 3.15 ± 0.10° to 3.46 ± 0.05° (*p* < 0.05) and from 1.35 ± 0.28° to 1.69 ± 0.30° (*p* < 0.001), respectively. In addition, PPH had abduction that increased to 1.91 ± with 9 cm heels (*p* < 0.001). On the other hand, I MTT had a valgus decrease with 6 cm heels (from 3.94 ± 0.28 to 3.70 ± 0.13 (*p* < 0.001)) but without any concordance correlation.

Finally, Table 6 and Table 7 summarize the Spearman’s Rho correlation coefficients of static and dynamic conditions between the height of the heels and the movements of PPH and I MTT, respectively.

In static tests, adduction of IMTT had a statistically significant positive correlation while wearing 6 cm heels versus unshod condition (0.5, *p* < 0.001) as well as abduction (0.347, *p* < 0.001). Valgus values had a statistically significant positive correlation too under 3 cm high heel (0.51, *p* < 0.001). Regarding PPH abduction, there was a statistical positive correlation in 6 cm heels (0.261, *p* < 0.05) as well as a statistically significant negative correlation to adduction under 3 cm and 6 cm of high heels (*p* < 0.05).

During dynamic tests, PPH abduction had a statistically significant positive correlation with heels of 6 cm (0.527, *p* < 0.001) combined with a statistically significant inverse correlation in adduction (−0.278, *p* < 0.05) with 6 cm; there was a significant positive correlation with valgus values (0.242, *p* < 0.05).

## 4. Discussion

HAV is a forefoot pathology related to PPH in valgus and abduction deviations plus the IMTT bone in adduction deviation [1,2,51]. Previous studies [18,52,53,54] have shown the relationship between HAV development and the narrow toe tip footwear typical of the high heel shoes; other groups [1,27] speculated that high heels have an etiology factor in HAV development with only one prior review [12] considering the isolated high heels’s effect on first MPJ deviation on its conclusions but without any further study. This research was the first study to use the Polhemus Fastrack^®^ to assess the effects of high heels on static and dynamic conditions on PPH and IMTT. Thus, many statistically significant variables have been obtained, but of the 96 variables studied here across 80 subjects, only a few had statistically significant correlations. Due to the independent evaluation of the bones, we could not establish a direct discussion with the findings of other authors who interpreted HAV as a global forefoot disease [34]. In addition, this work has studied a healthy population without any limitation of mobility in their joints. Thus, a small difference in segment movements was expected.

Our results on the effect of high heels on PPH during dynamic testing showed that a high heel of 6 cm had a statistically significant increase in abduction in the transverse plane and valgus movement in the frontal plane through IMPJ. There was positive correlation of these values as well as a statistically significant reduction in the adduction; there was negative correlation to the barefoot condition. In addition, during static tests, there was a statistically significant abduction increase in PPH for high heels 6 cm or higher with a corresponding positive correlation; therefore, abduction is the only movement that appeared to have a positive correlation and statistical significance in both static and dynamic tests. Thus, we conclude that heels over 6 cm correlated with an abduction effect on PPH without the narrow shoe box interference. This agrees with arguments on the biomechanical development of HAV processes that claimed that the PPH was the first precursor bone segment to begin the HAV pathology [51,55,56,57] due to medial capsular tension ligaments that become hyper-elastic and let the PPH proceed to abduction deviation [7]. The PPH then has a strong push forward to the IMTT in the push off phase that can lead to adduction deviation.

More recently, Wang et al. [34] reported an increase in forefoot abduction while wearing 5 cm high-heeled shoes during walking vs. barefoot arguing that the squeezing effect of the high heels on the foot had a displacement toward the toe tip. This produced valgus and abduction of PPH. We agree with these conclusions, but we showed that the foot produces this “abductor effect” on PPH; it is not from the narrow box of the shoes. In addition, we obtained a large increase in the value of PPH abduction with a 3 cm heel. This result was not statistically significant.

The effects of high heels had contradictory effects on IMTT movements. There were no statistically significant results to justify its implication on HAV development in contrast with other groups that identified IMTT adduction and valgus [1,2] as well as inclination of the IMTT axis as risk factors of bunion [58] or IMCJ hypermobility [59]; both movements on transverse and frontal planes are under doubt because of a lack of objective data [60]. Surprisingly, it seemed that wearing any high heel might decrease valgus deviation of IMTT although this condition only had statistical significance with 6 cm heels; there was no positive correlation. The absence of concrete IMTT values related with typical HAV development suggests that PPH may be the principal bone to start the pathological process. This agrees with a study that identified the presence of HAV with greater reduction in size of the adductor hallucis muscle [61] as one of the most important muscles to balance the PPH.

In contrast to previous studies [62] that found no association between footwear characteristics (heel height and narrow box) and HAV development, our cohort had (18–38 years) had movement deviation of PPH in heels over 6 cm. This agrees with other studies where older women reported HAV. They wore shoes with heels over 5 cm [12]. This work showed data on IMTT and PPH from the two different static and dynamic conditions and 3 kinds of high heels. Thus, we selected and summarized heel heights and determined that high heels could develop HAV. 6 cm was the common height for both static and dynamic situations; these different variables converged to induce HAV development.

Most of the main limitations in other similar studies were equivocal results secondary to small sample size [19], differences in anthropometric characteristics of the subjects groups [63], or the inclusion of participants wearing their own high-heeled shoes [64]. This leads to a heterogeneous sample [19]. We studied a homogeneous sample that improved the measurement conditions of other groups that also failed to show ICC, CCC, SEM or MDC values [10,34]; our data had low to moderate correlation, and we considered these statistical parameters.

Coupling relationships between hindfoot inversion/eversion and forefoot abduction/adduction (*R*^2^ = 0.5) and hallux dorsiflexion/plantarflexion (*R*^2^ = 0.7) were the only prior references found on this topic [65]. It had similar Spearman values in that work but did show any contrast between data because the authors did not assess individual movements of any segment bone as PPH or I MTT, like in the present work. These are the only specific correlations done relative to PPH and IMTT in the literature and confirm the low Spearman correlations of our values.

## 5. Limitations

Bone segment measurements were performed with an electronic Polhemus Fastrack^®^ goniometer and showed current instability due to inherent human gait fluctuations and sensor noise. The MDC values were higher than the grades obtained in the dynamic PPH valgus and abduction as well as in the static abduction PPH variables. Considering that the percent error set for the device was around 1.6%—and considering that the ±SD and SEM values obtained in the study were under this 1.6%—these results are considered to be statistically valid but with caution.

On the other hand, we were not able to study the effect of order on our sample because didn’t write the different orders of each one neither the number of these selections. We assumed the possible “order effect” as “perfectly balanced” because all the study subgroups of each station have the same number of subjects and this can dilutes the “order effect”. Future study design should include a section of studying the effect of order of experiments.

## 6. Conclusions

Wearing shoes with heels over 6 cm may produce a valgus and abduction increase in PPH movement. This abduction is specifically detected in the development of HAV pathology. Future cohort studies will be required to clarify the time period that is needed to develop HAV pathology related to high-heeled shoes.

## Figures and Tables

**Figure 1 sensors-19-01328-f001:**
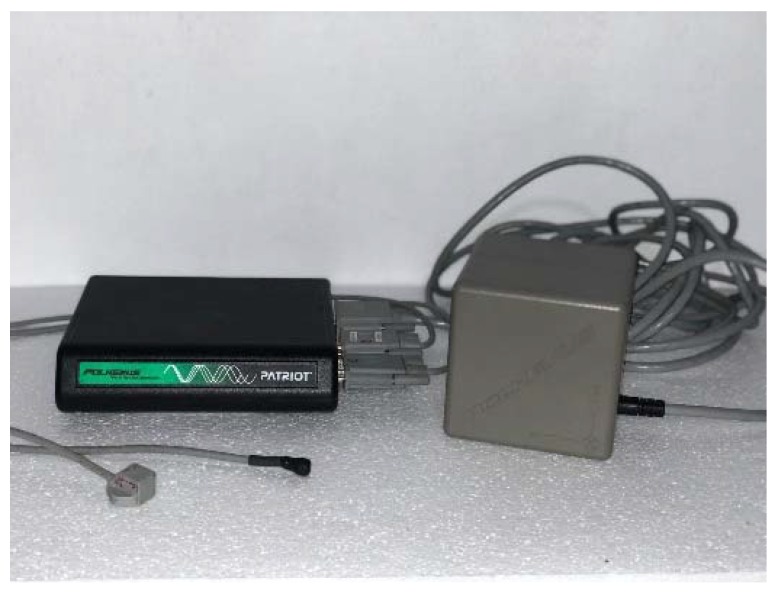
Polhemus device. From left to the right: sensors, receiver module, emitter module.

**Figure 2 sensors-19-01328-f002:**
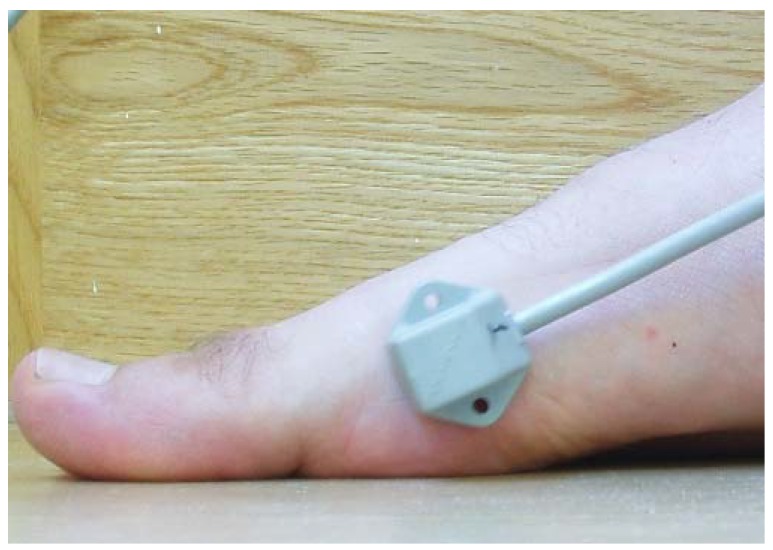
Sensor one. Location placed on the medial head of first metatarsal bone.

**Figure 3 sensors-19-01328-f003:**
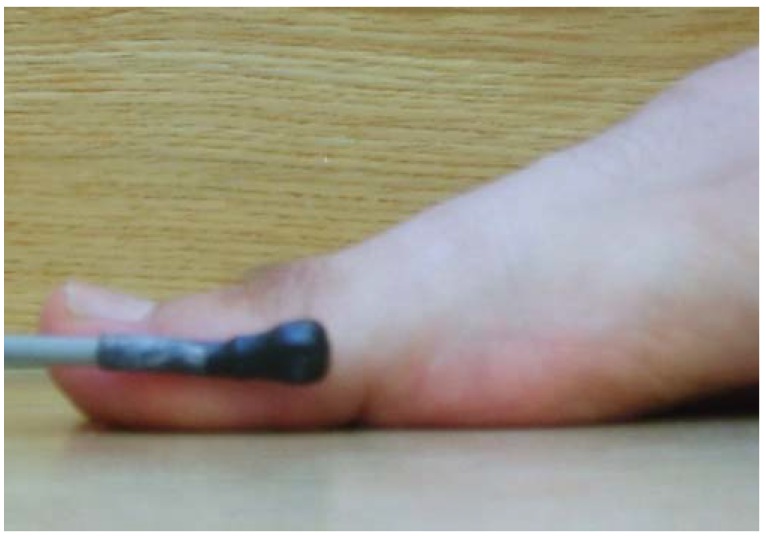
Sensor two. Location placed on the border of the proximal phalanx of the hallux.

**Figure 4 sensors-19-01328-f004:**
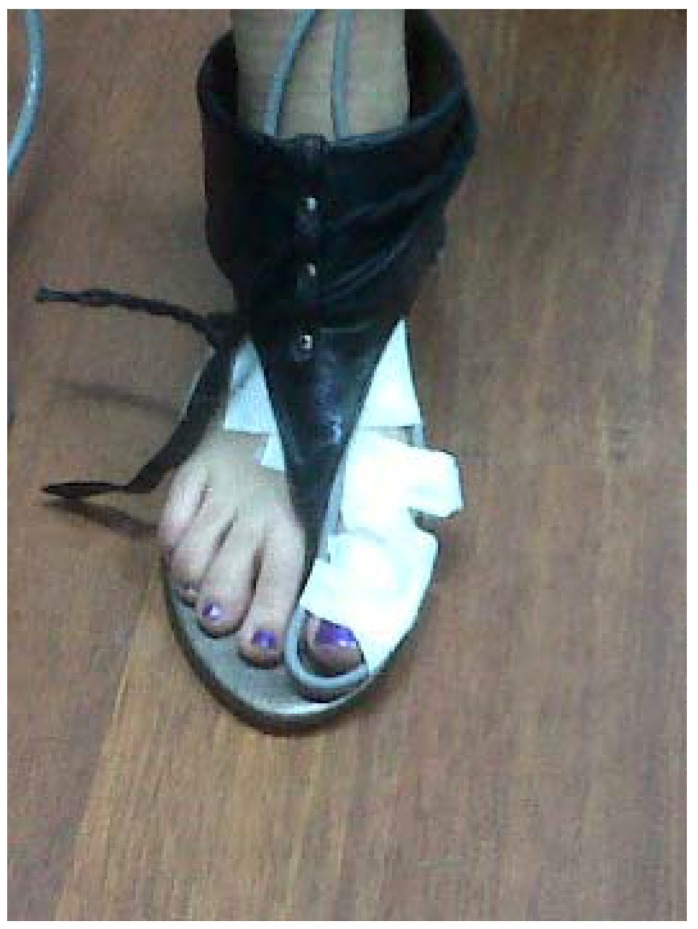
Fastening of the sensors. White bandages are auto-adhesive fixatives.

**Figure 5 sensors-19-01328-f005:**
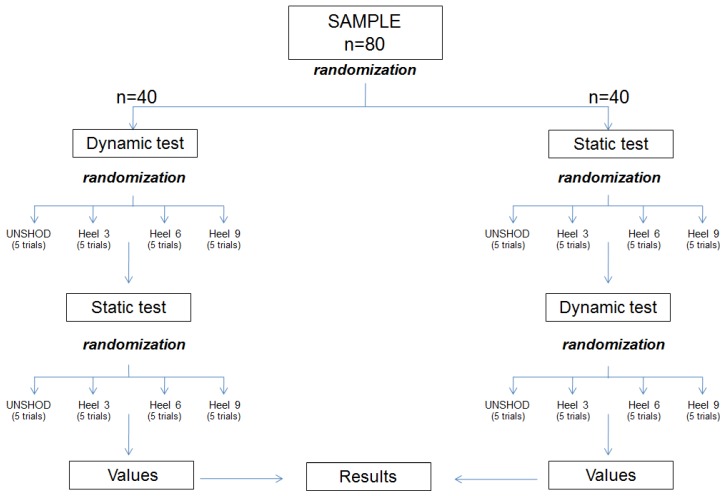
Randomized flow chart.

**Figure 6 sensors-19-01328-f006:**
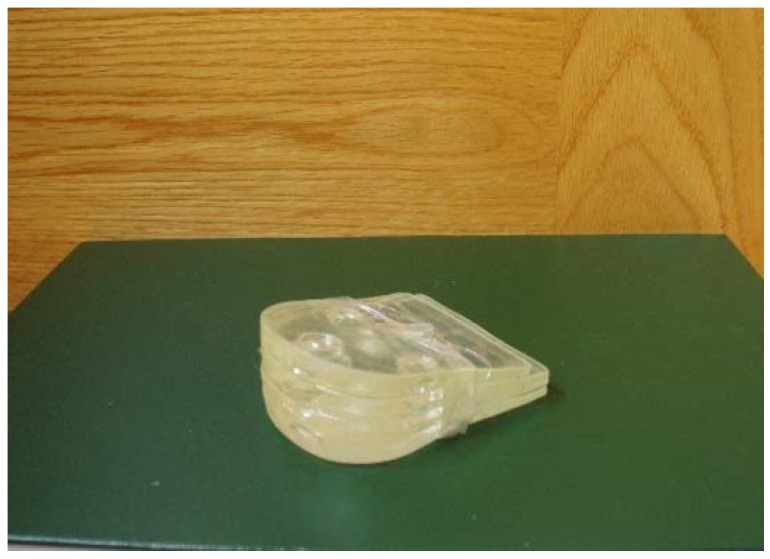
Prefabricated high-heels. The heels were made on hard silicone. The additional height was added over the top.

**Figure 7 sensors-19-01328-f007:**
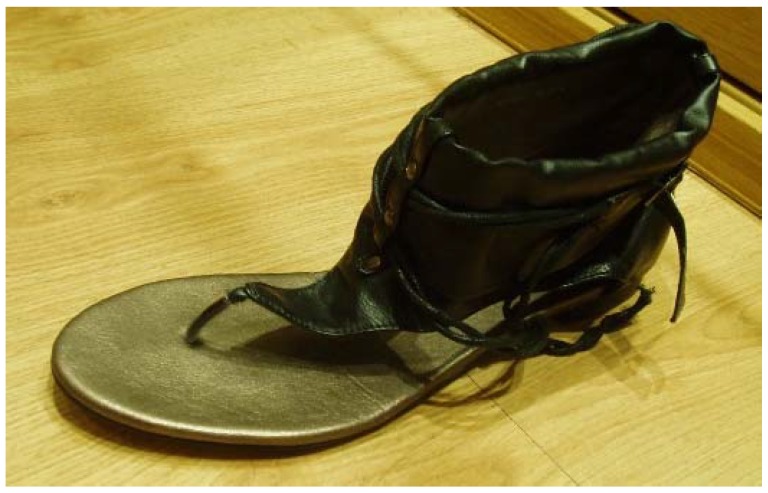
Sandal model used in the study. A free toe-tip-box is present to allow movement of proximal phalanx of the hallux and first metatarsal bone.

**Figure 8 sensors-19-01328-f008:**
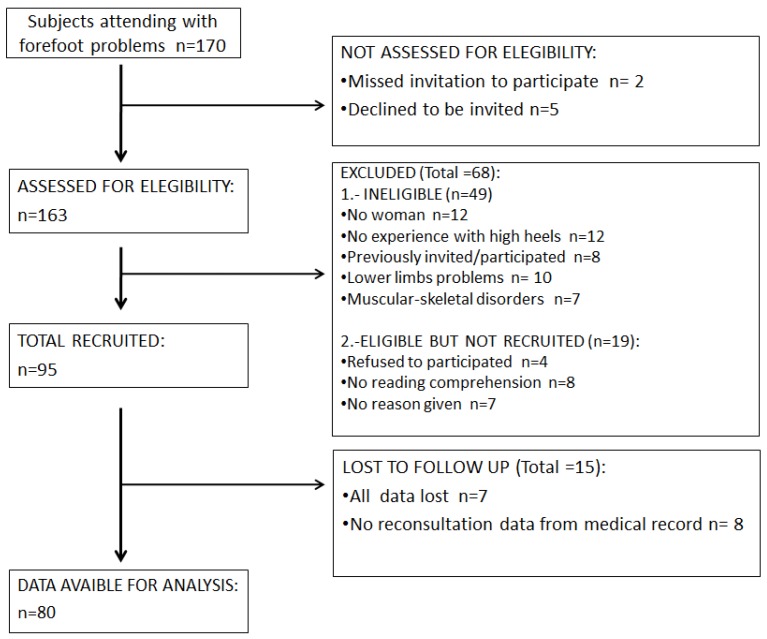
Participant flow chart.

**Table 1 sensors-19-01328-t001:** Sociodemographic characteristics of the participants.

Variable	Female *n* = 80
	Mean ± SD (95% CI)
Age (years)	25.1 ± 4.67 (26.11–24.08)
Height (cm)	164.03 ± 5.44 (165.2–162.85)
Weight (kg)	57.53 ± 6.304 (58.90–56.15)
Foot Size (EC)	38.13 ± 1.184 (38.37–37.88)
BMI (kg/m^2^)	20.2 ± 1.74 (20.58–19.82)

Abbreviations: EC = European countries; BMI = body mass index; SD = Standard Deviation; CI = Confidence Interval.

**Table 2 sensors-19-01328-t002:** Reliability ICC and CCC of static variables in unshod versus heels that were 3 cm, 6 cm, or 9 cm high.

Variable	Unshod	Heel 3 cm	Unshod-Heel 3 cm			Heel 6 cm	Unshod-Heel 6 cm			Heel 9 cm	Unshod-Heel 9 cm		
	ICC (95%CI)	ICC (95%CI)	CCC (95%CI)	SEM	MDC	ICC (95%CI)	CCC (95%CI)	SEM	MDC	ICC (95%CI)	CCC (95%CI)	SEM	MDC
					95%				95%				95%
MTT-AD	0.986	0.968	0.612	1.14	3.161	0.991	0.477	1.275	3.533	0.984	0.184	1.455	4.034
	(0.981–0.991)	(0.983–0.991)	(0.493–0.7) **			(0.988–0.994)	(0.341–0.594) **			(0.978–0.0989)	(0.106–0.26) **		
MTT-AB	0.904	0.936	0.15	0.64	1.774	0.985	0.258	0.598	1.658	0.961	0.136	0.711	1.97
	(0.866–0.933)	(0.91–0.956)	(0.04–0.33) **			(0.965–0.862)	(0.042–0.45)			(0.945–0.973)	(0.012–0.277) **		
PPH-AD	0.981	0.95	0.096	1.357	3.762	0.969	0.054	1.65	4.574	0.944	0.147	1.298	3.598
	(0.964–0.992)	(0.93–0.965)	(0.039–0.153) **			(0.956–0.978)	(0.11–0.13) **			(0.922–0.961)	(0.081–0.212)		
PPH-ABD	0.94	0.908	0.006	0.828	2.296	0.926	0.239	0.591	1.639	0.969	0.168	0.63	1.747
	(0.916–0.958)	(0.872–0.937)	(0.01–0.019) **			(0.897–0.949)	(0.040–0.41) **			(0.957–0.979)	(0.044–0.365) **		
MTT-PF	0.997	0.994	0.792	2.493	6.911	0.991	0.162	5.633	15.615	0.99	0.19	5.381	14.915
	(0.995–0.998)	(0.992–0.996)	(0.727–0.842)			(0.988–0.994)	(0.033–0.286)			(0.986–0.993)	(0.104–0.272) *		
MTT-DF	0.902	0.928	0.414	0.304	0.844	0.918	0.487	0.27	0.749	0.968	0.107	0.579	1.604
	(0.864–0.932)	(0.90–0.95)	(0.219–0.577)			(0.886–0.944)	(0.30–0.63)			(0.955–0.978)	(0.008–0.218) *		
PPH-PF	0.983	0.982	0.034	1.825	5.059	0.988	0.091	1.67	4.63	0.979	0.19	1.574	4.363
	(0.977–0.989)	(0.974–0.987)	(0.16–0.23)			(0.984–0.992)	(0.10–0.279)			(0.971–0.986)	(0.013–0.356) **		
PPH-DF	0.956	0.978	0.381	1.213	3.361	0.99	0.26	1.628	4.512	0.982	0.153	2.274	6.303
	(0.939–0.970)	(0.969–0.984)	(0.182–0.549)			(0.986–0.993)	(0.066–0.435)			(0.975–0.988)	(0.037–0.67) **		
MTT-VL	0.975	0.994	0.043	2.471	6.848	0.961	0.158	1.202	3.333	0.973	0.082	1.277	3.539
	(0.965–0.983)	(0.992–0.996)	(0.004–0.091) **			(0.946–0.983)	(0.069–0.244) **			(0.962–0.981)	(0.033–0.13) **		
MTT-VR	0.995	0.976	0.178	2.419	6.706	0.99	0.64	1.463	4.054	0.971	0.43	1.463	4.747
	(0.993–0.996)	(0.967–0.984)	(0.29–0.31) *			(0.986–0.993)	(0.513–0.739) **			(0.960–0.98)	(0.33–0.52) **		
PPH-VL	0.989	0.993	0.455	2.023	5.608	0.985	0.162	2.336	6.476	0.981	0.128	2.143	5.94
	(0.985–0.993)	(0.99–0.995)	(0.328–0.567) **			(0.979–0.990)	(0.035–0.347) **			(0.974–0.987)	(0.051–0.299) **		
PPH-VR	0.98	0.978	0.388	1.395	3.866	0.975	0.152	1.921	5.326	0.989	0.13	2.851	7.902
	(0.972–0.986)	(0.969–0.985)	(0.198–0.55)			(0.966–0.983)	(0.058–0.35) **			(0.984–0.992)	(0.047–0.18) **		

Abbreviations: ICC = Intraclass Correlation Coefficient; CI = Confidence Interval; CCC = Concordance Correlation Coefficient; SEM = standard error of measurement; MDC = minimal detectable change; MTT ADD = first metatarsal bone in Adduction, I MTT toward to medial body line; MTT ABD = first metatarsal bone in Abduction, I MTT away to medial body line; MTT PF = first metatarsal bone in plantarflexion, I MTT toward to the floor; MTT DF = first metatarsal bone in dorsiflexion, I MTT away to the floor; MTT VL = first metatarsal bone spins to inside, toward medial body line; MTT VR = first metatarsal bone spin to outside, away to medial body line; PPH AD = proximal phalanx of hallux in adduction: PPH toward to medial body line; PPH ABD = proximal phalanx of hallux in abduction: PPH away to medial body line; PPH PF = proximal phalanx of hallux in plantarflexion: PPH toward to the floor; PPH DF = proximal phalanx of hallux in dorsiflexion: PPH away to the floor; PPH VL = proximal phalanx of hallux spins to inside: PPH toward medial body line; PPH VR = proximal phalanx of hallux spin to outside: PPH away to medial body line; UNSHOD = barefoot participants without any height in shoe heel; ±SD = standard deviation. Movements and axes refer to the medial line of the body; all values are expressed in degrees; * *p*-value < 0.05; ** *p*-value < 0.001.

**Table 3 sensors-19-01328-t003:** Reliability ICC and CCC of dynamic variables in unshod versus heels that were 3 cm, 6 cm, or 9 cm high.

Variable	Unshod	Heel 3 cm	Unshod-Heel 3 cm			Heel 6 cm	Unshod-Heel 6 cm			Heel 9 cm	Unshod-Heel 9 cm		
	ICC (95%CI)	ICC (95%CI)	CCC (95%CI)	SEM	MDC	ICC (95%CI)	CCC (95%CI)	SEM	MDC	ICC (95%CI)	CCC (95%CI)	SEM	MDC
					95%				95%				95%
MTT-AD	0.988	0.986	0.191	1.953	5.414	0.968	0.25	1.785	4.949	0.986	0.232	1.813	5.026
	(0.983–0.992)	(0.981–0.990)	(0.123–0.25) **			(0.955–0.978)	(0.171–0.327) **			(0.981–0.99)	(0.15–0.30)		
MTT-AB	0.993	0.993	0.041	3.529	9.782	0.997	0.151	3.499	9.699	0.992	0.28	2.715	7.526
	(0.991–0.995)	(0.990–0.995)	(0.48–0.131)			(0.995–0.998)	(0.055–0.245) *			(0.988–0.944)	(0.18–0.46)		
PPH-AD	0.938	0.958	0.049	1.051	2.914	0.965	0.026	1.334	3.698	0.977	0.197	1.117	3.096
	(0.913–0.957)	(0.941–0.971)	(0.11–0.17) **			(0.951–0.976)	(0.10–0.157) **			(0.969–0.984)	(0.06–0.321) *		
PPH-ABD	0.976	0.983	0.04	1.852	5.132	0.981	0.51	0.782	2.169	0.994	0.295	1.793	4.971
	(0.966–0.983)	(0.976–0.988)	(0.03–0.11) **			(0.973–0.987)	(0.33–0.65) **			(0.991–0.996)	(0.17–0.41) **		
MTT-PF	0.998	0.997	0.834	4.421	12.255	0.998	0.799	5.051	14.001	0.992	0.268	9.662	26.782
	(0.998–0.999)	(0.996–0.998)	(0.77–0.88)			(0.997–0.999)	(0.714–0.861)			(0.988–0.994)	(0.14–0.38)		
MTT-DF	0.935	0.898	0.356	0.404	1.121	0.933	0.198	0.416	1.153	0.953	0.453	0.417	1.155
	(0.910–0.955)	(0.857–0.929)	(0.186–0.521)			(0.907–0.954)	(0.015–0.368)			(0.935–0.968)	(0.262–0.61)		
PPH-PF	0.999	0.999	0.723	6.755	18.723	0.998	0.857	4.929	13.662	0.996	0.425	9.801	27.166
	(0.999–1.00)	(0.998–0.999)	(0.608–0.808)			(0.998–0.999)	(0.787–0.9)			(0.995–0.997)	(0.298–0.538)		
PPH-DF	0.981	0.917	0.175	1.181	3.272	0.928	0.198	1.17	3.243	0.958	0.231	1.195	3.313
	(0.974–0.987)	(0.844–0.943)	(0.002–0.185)			(0.9–0.950)	(0.015–0.36)			(0.941–0.971)	(0.038–0.40) *		
MTT-VL	0.985	0.99	0.137	2.326	6.446	0.994	0.075	2.997	8.306	0.993	0.12	2.565	7.109
	(0.979–0.990)	(0.986–0.993)	(0.020–0.250) **			(0.992–0.996)	(0.05–0.2) **			(0.991–0.995)	(0.017–0.21) **		
MTT-VR	0.985	0.982	0.26	2.314	6.413	0.947	0.15	2.476	6.864	0.928	0.042	2.703	7.491
	(0.979–0.989)	(0.975–0.988)	(0.183–0.334) **			(0.927–0.964)	(0.084–0.214)			(0.9–0.951)	(0.04–0.05) *		
PPH-VL	0.964	0.99	0.392	1.928	5.343	0.986	0.225	2.022	5.605	0.969	0.12	2.439	6.76
	(0.950–0.975)	(0.986–0.993)	(0.293–0.484) **			(0.981–0.99)	(0.131–0.315) **			(0.956–0.978)	(0.011–0.246) *		
PPH-VR	0.992	0.987	0.53	1.535	4.256	0.951	0.384	1.577	4.37	0.981	0.472	1.656	4.591
	(0.989–0.995)	(0.981–0.991)	(0.364–0.664) **			(0.931–0.966)	(0.265–0.491) **			(0.9–0.951)	(0.291–0.621)		

Abbreviations: ICC = Intraclass Correlation Coefficient; CI = Confidence Interval; CCC = Concordance Correlation Coefficient; SEM = standard error of measurement; MDC = minimal detectable change; MTT ADD = first metatarsal bone in Adduction, I MTT toward to medial body line; MTT ABD = first metatarsal bone in Abduction, I MTT away to medial body line; MTT PF = first metatarsal bone in plantarflexion, I MTT toward to the floor; MTT DF = first metatarsal bone in dorsiflexion, I MTT away to the floor; MTT VL = first metatarsal bone spins to inside, toward medial body line; MTT VR = first metatarsal bone spin to outside, away to medial body line; PPH AD = proximal phalanx of hallux in adduction: PPH toward to medial body line; PPH ABD = proximal phalanx of hallux in abduction: PPH away to medial body line; PPH PF = proximal phalanx of hallux in plantarflexion: PPH toward to the floor; PPH DF = proximal phalanx of hallux in dorsiflexion: PPH away to the floor; PPH VL = proximal phalanx of hallux spins to inside: PPH toward medial body line; PPH VR = proximal phalanx of hallux spin to outside: PPH away to medial body line; UNSHOD = barefoot participants without any height in shoe heel; ±SD = standard deviation. Movements and axes refer to the medial line of the body; all values are expressed in degrees; * *p*-value < 0.05; ** *p*-value < 0.001.

**Table 4 sensors-19-01328-t004:** STATIC mobility grades of first metatarsal (sensor 1) and proximal phalanx of hallux (sensor 2) bones in unshod with heels of 3 cm, 6 cm, and 9 cm high.

Variable	Unshod	Heel 3 cm	Heel 6 cm	Heel 9 cm	*p*-Value	*p*-Value	*p*-Value
	Mean (°) ± SD (95% CI)	Mean (°) ± SD (95% CI)	Mean (°) ± SD (95% CI)	Mean (°) ± SD (95% CI)	Unshod vs. Heel 3 cm	Unshod vs. Heel 6 cm	Unshod vs. Heel 9 cm
MTT AD	3.78 ± 0.36	2.14 ± 0.41	2.04 ± 0.48	1.20 ± 0.25	<0.001 **	<0.001 **	<0.001 **
	(3.70–3.85)	(2.05–2.22)	(1.93–2.14)	(1.14–1.25)			
MTT AB	1.32 ± 0.16	1.74 ± 0.12	1.36 ± 0.17	2.14 ± 0.17	1.115	<0.001 **	<0.05 *
	(1.28–1.35)	(1.71–1.76)	(1.32–1.39)	(2.10–2.17)			
MTT PF	23.88 ± 1.90	24.12 ± 1.00	26.69 ± 1.25	17.68 ± 0.61	<0.001 **	<0.05 *	<0.001 **
	(23.46–24.29)	(23.90–24.33)	(26.41–26.96)	(17.54–17.81)			
MTT DF	0.66 ± 0.11	0.57 ± 0.10	0.62 ± 0.10	1.45 ± 0.17	<0.001 **	<0.001 **	0.064
	(0.63–0.68)	(0.54–0.59)	(0.59–0.64)	(1.41–1.48)			
MTT VL	1.77 ± 0.20	2.15 ± 0.10	4.00 ± 0.24	3.46 ± 0.15	<0.001 **	0.501	0.062
	(1.72–1.81)	(2.12–2.17)	(3.94–4.05)	(3.42–3.49)			
MTT VR	1.50 ± 0.23	3.87 ± 0.20	0.93 ± 0.19	1.76 ± 0.18	<0.001 **	<0.001 **	<0.001 **
	(1.44–1.55)	(3.82–3.91)	(0.88–0.97)	(1.72–1.79)			
PPH ADD	1.34 ± 0.09	1.60 ± 0.13	1.05 ± 0.12	1.78 ± 0.15	0.239	0.268	0.216
	(1.32–1.35)	(1.57–1.62)	(1.02–1.07)	(1.74–1.81)			
PPH ABD	1.01 ± 0.36	2.32 ± 0.18	1.31 ± 0.46	1.17 ± 0.15	0.907	<0.05 *	0.121
	(0.93–1.08)	(2.28–2.35)	(1.20–1.41)	(1.13–1.20)			
PPH PF	3.32 ± 0.29	3.90 ± 0.35	4.37 ± 0.42	4.64 ± 0.43		0.893	0.642
	(3.25–3.38)	(3.82–3.97)	(4.27–4.46)	(4.54–4.73)	<0.05 *		
PPH DF	2.23 ± 0.37	2.40 ± 0.43	1.94 ± 0.55	5.79 ± 0.48	<0.001 **	<0.05 *	<0.001 **
	(2.14–2.31)	(2.30–2.49)	(1.81–2.06)	(5.68–5.89)			
PPH VL	2.57 ± 0.12	2.31 ± 0.12	3.07 ± 0.16	3.52 ± 0.09	0.555	0.102	0.419
	(2.54–2.59)	(2.28–2.33)	(3.03–3.10)	(3.50–3.53)			
PPH VR	1.89 ± 0.22	2.70 ± 0.19	1.29 ± 0.18	1.78 ± 0.15	<0.05 *	<0.05 *	0.169
	(1.84–1.93)	(2.65–2.74)	(1.25–1.32)	(1.74–1.81)			

Abbreviations: MTT ADD = first metatarsal bone in Adduction: I MTT toward to medial body line; MTT ABD = first metatarsal bone in abduction: I MTT away to medial body line; MTT PF = first metatarsal bone in plantarflexion: I MTT toward to the floor; MTT DF = first metatarsal bone in dorsiflexion: I MTT away to the floor; MTT VL = first metatarsal bone spins to inside: I MTT toward medial body line; MTT VR = first metatarsal bone spin to outside: I MTT away to medial body line; UNSHOD = barefoot participants without any height in shoe heel; PPH ADD = proximal phalanx of hallux in adduction: PPH toward to medial body line; PPH ABD = proximal phalanx of hallux in Abduction: PPH away to medial body line; PPH PF = proximal phalanx of hallux in plantarflexion: PPH toward to the floor; PPH DF = proximal phalanx of hallux in dorsiflexion: PPH away to the floor; PPH VL = proximal phalanx of hallux spins to inside: PPH toward medial body line; PPH VR = proximal phalanx of hallux spin to outside: PPH away to medial body line; ±SD = standard deviation; *p-*value < 0.05 ***** (within a 95% confidence interval) was considered statistically significant; *p-*value < 0.001 ** (within a 95% confidence interval) was considered statistically strong significant; Movements and axes refer to the medial line of the body; all values are expressed in degrees.

**Table 5 sensors-19-01328-t005:** DYNAMIC mobility grades of first metatarsal (sensor 1) and proximal phalanx of hallux (sensor 2) bones in unshod as well as heels 3 cm, 6 cm and 9 cm high.

Variable	Unshod	Heel 3 cm	Heel 6 cm	Heel 9 cm	*p-*Value	*p-*Value	*p-*Value
	Mean (°) ± SD (95% CI)	Mean (°) ± SD (95% CI)	Mean (°) ± SD (95% CI)	Mean (°) ± SD (95% CI)	Unshod vs. Heel 3 cm	Unshod vs. Heel 6 cm	Unshod vs. Heel 9 cm
MTT ADD	4.97 ± 0.36	1.42 ± 0.35	2.51 ± 0.30	2.25 ± 030	0.226	0.088	0.852
	(4.89–5.04)	(1.34–1.49)	(2.43–2.59)	(2.17–2.32)			
MTT ABD	1.89 ± 0.17	0.76 ± 0.10	1.04 ± 0.08	2.01 ± 0.14	0.211	<0.001 **	<0.05 *
	(1.85–1.92)	(0.73–0.78)	(1.02–1.06)	(1.96–2.04)			
MTT PF	38.30 ± 3.38	35.06 ± 1.99	36.13 ± 2.67	44.88 ± 1.40	<0.001 **	<0.001 **	<0.05 *
	(37.55–39.04)	(34.62–35.49)	(35.54–36.71)	(44.57–45.18)			
MTT DF	0.97 ± 0.15	0.83 ± 0.10	0.86 ± 0.13	0.81 ± 0.14	<0.001 **	<0.001 **	<0.001 **
	(0.93–1.00)	(0.80–0.85)	(0.83–0.89)	(0.76–0.86)			
MTT VL	3.94 ± 0.28	2.88 ± 0.10	3.70 ± 0.13	3.60 ± 0.08	0.724	<0.001 **	0.526
	(3.87–4.00)	(2.85–2.90)	(3.67–3.73)	(3.58–3.61)			
MTT VR	3.92 ± 0.19	1.11 ± 0.21	1.08 ± 0.15	2.27 ± 0.26	0.98	0.25	0.653
	(3.85–3.94)	(1.06–1.16)	(1.04–1.11)	(2.21–2.32)			
PPH ADD	2.39 ± 0.13	0.87 ± 0.15	1.31 ± 0.14	1.12 ± 0.19	<0.001 **	0.088	<0.001 **
	(2.36–2.41)	(0.83–0.90)	(1.26–1.33)	(1.07–1.16)			
PPH ABD	1.35 ± 0.28	4.56 ± 0.50	1.69 ± 0.30	1.91 ± 0.24	0.486	<0.001 **	<0.05 *
	(1.28–1.41)	(4.45–4.66)	(1.66–1.71)	(1.85–1.96)			
PPH PF	8.89 ± 0.48	4.66 ± 0.10	8.87 ± 0.48	2.25 ± 0.21	0.372	<0.001 **	0.605
	(8.78–8.99)	(4.63–4.69)	(8.76–8.97)	(2.20–2.29)			
PPH DF	1.97 ± 0.39	1.67 ± 0.20	1.97 ± 0.22	1.56 ± 0.27	0.051	<0.05 *	<0.05 *
	(1.88–2.05)	(1.62–1.71)	(1.92–2.01)	(1.50–1.63)			
PPH VL	3.15 ± 0.10	3 ± 0.11	3.46 ± 0.05	4.31 ± 0.04	0.411	<0.05 *	0.873
	(3.12–3.17)	(2.97–3.02)	(3.45–3.46)	(4.30–4.31)			
PPH VR	1.56 ± 0.33	1.26 ± 0.27	1.26 ± 0.22	1.68 ± 0.29	<0.001 **	<0.001 **	<0.05 *
	(1.48–1.63)	(1.20–1.31)	(1.21–1.30)	(1.61–1.74)			

Abbreviations: MTT ADD = first metatarsal bone in Adduction: I MTT toward to medial body line; MTT ABD = first metatarsal bone in Abduction: I MTT away to medial body line; MTT PF = first metatarsal bone in plantarflexión: I MTT toward to the floor; MTT DF = first metatarsal bone in dorsiflexión: I MTT away to the floor; MTT VL = first metatarsal bone spins to inside: I MTT toward medial body line; MTT VR = first metatarsal bone spin to outside: I MTT away to medial body line; UNSHOD = barefoot participants without any height in shoe heel; PPH ADD = proximal phalanx of hallux in Adduction: PPH toward to medial body line; PPH ABD = proximal phalanx of hallux in Abduction: PPH away to medial body line; PPH PF = proximal phalanx of hallux in plantarflexión: PPH toward to the floor; PPH DF = proximal phalanx of hallux in dorsiflexión: PPH away to the floor; PPH VL = proximal phalanx of hallux spins to inside: PPH toward medial body line; PPH VR = proximal phalanx of hallux spin to outside: PPH away to medial body line; ±SD = standard deviation; *p-*value < 0.05 * (within a 95% confidence interval) was considered statistically significant; *p-*value < 0.001 ** (within a 95% confidence interval) was considered statistically strong significant; Movements and axes refer to the medial line of the body; all values are expressed in degrees.

**Table 6 sensors-19-01328-t006:** STATIC Spearman-Rho correlation coefficients between variables.

Variable	Unshod vs. Heel 3 cm	*p-*Value	Unshod vs. Heel 6 cm	*p-*Value	Unshod vs. Heel 9 cm	*p-*Value
	(ρ-Coefficient)		(ρ-Coefficient)		(ρ-Coefficient)	
MTT ADD	0.256	0.3	0.5	<0.001 **	0.521	<0.001 **
MTT ABD	−0.69	0.77	0.347	<0.001 **	0.228	<0.05 *
MTT PF	−0.189	−0.192	0.723	0.061	−0.125	0.75
MTT DF	0.173	0.124	−0.639	0.072	0.146	0.6
MTT VL	0.510	<0.001 **	−0.370	0.5	0.487	0.336
MTT VR	0.26	0.122	−0.401	<0.001 **	0.587	0.2
PPH ADD	0.391	0.46	0.150	−0.112	0.0984	0.226
PPH ABD	0.412	0.477	0.261	<0.05 *	0.58	0.398
PPH PF	0.184	0.103	−0.017	0.116	−0.189	0.69
PPH DF	0.054	0.111	−0.079	0.086	−0.141	0.333
PPH VL	0.179	−0.037	0.71	0.406	0.183	522
PPH VR	−0.306	<0.05 *	−0.222	<0.05 *	0.36	0.922

Abbreviations: ρ-coefficient = Spearman Rho coefficient; MTT ADD = first metatarsal bone in adduction: I MTT toward to medial body line; MTT ABD = first metatarsal bone in abduction: I MTT away to medial body line; MTT PF = first metatarsal bone in plantarflexion: I MTT toward to the floor; MTT DF = first metatarsal bone in dorsiflexion: I MTT away to the floor; MTT VL = first metatarsal bone spins to inside: I MTT toward medial body line; MTT VR = first metatarsal bone spin to outside: I MTT away to medial body line; UNSHOD = barefoot participants without any height in shoe heel; PPH ADD = proximal phalanx of hallux in adduction: PPH toward to medial body line; PPH ABD = proximal phalanx of hallux in abduction: PPH away to medial body line; PPH PF = proximal phalanx of hallux in plantarflexion: PPH toward to the floor; PPH DF = proximal phalanx of hallux in dorsiflexion: PPH away to the floor; PPH VL = proximal phalanx of hallux spins to inside: PPH toward medial body line; PPH VR = proximal phalanx of hallux spin to outside: PPH away to medial body line; ±SD = standard deviation; *p*-value < 0.05 * (within a 95% confidence interval) was considered statistically significant; *p*-value < 0.001 ** (within a 95% confidence interval) was considered statistically strong significant; Movements and axes refer to the medial line of the body; all values are expressed in degrees.

**Table 7 sensors-19-01328-t007:** DINADYNAMIC Spearman-Rho correlation coefficients between variables.

Variable	Unshod vs. Heel 3 cm	*p-*Value	Unshod vs. Heel 6 cm	*p-*Value	Unshod vs. Heel 9 cm	*p-*Value
	(ρ-Coefficient)		(ρ-Coefficient)		(ρ-Coefficient)	
MTT ADD	−0.276	<0.05 *	0.025	0.061	0.181	0.3
MTT ABD	0.306	0.189	0.606	0.138	0.258	0.224
MTT PF	0.086	−0.193	0.026	−0.139	−0.119	0.148
MTT DF	0.056	0.203	0.043	0.128	−0.142	0.956
MTT VL	0.167	0.128	−0.104	0.134	0.523	<0.001 **
MTT VR	−0.071	0.0733	−0.088	0.0621	0.666	0.459
PPH ADD	−0.364	<0.001 **	−0.278	<0.05 *	0.856	0.1
PPH ABD	0.71	0.25	0.527	<0.001 **	0.375	0.476
PPH PF	−0.128	−0.005	−0.028	0.148	−0.109	0.945
PPH DF	0.1	−0.116	0.148	0.058	0.167	0.8
PPH VL	−0.169	0.150	0.242	<0.05 *	0.154	0.36
PPH VR	−0.522	<0.001 **	0.108	−0.089	0.146	0.221

Abbreviations: ρ-coefficient = Spearman Rho coefficient; MTT ADD = first metatarsal bone in Adduction: I MTT toward to medial body line; MTT ABD = first metatarsal bone in abduction: I MTT away to medial body line; MTT PF = first metatarsal bone in plantarflexion: I MTT toward to the floor; MTT DF = first metatarsal bone in dorsiflexion: I MTT away to the floor; MTT VL = first metatarsal bone spins to inside: I MTT toward medial body line; MTT VR = first metatarsal bone spin to outside: I MTT away to medial body line; UNSHOD = barefoot participants without any height in shoe heel; PPH ADD = proximal phalanx of hallux in adduction: PPH toward to medial body line; PPH ABD = proximal phalanx of hallux in abduction: PPH away to medial body line; PPH PF = proximal phalanx of hallux in plantarflexion: PPH toward to the floor; PPH DF = proximal phalanx of hallux in dorsiflexion: PPH away to the floor; PPH VL = proximal phalanx of hallux spins to inside: PPH toward medial body line; PPH VR = proximal phalanx of hallux spin to outside: PPH away to medial body line; ±SD = standard deviation; *p*-value < 0.05 * (within a 95% confidence interval) was considered statistically significant; *p*-value < 0.001 ** (within a 95% confidence interval) was considered statistically strong significant; Movements and axes refer to the medial line of the body; all values are expressed in degrees.

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
