# Peer review of "Heel Height as an Etiology of Hallux Abductus Valgus Development: An electromagnetic Static and Dynamic First Metatarsophalangeal Joint Study"

_sensors, 2019, doi:10.3390/s19061328_

Round 1
Reviewer 1 Report
Dear authors:
I have reviewed carefully the paper “Heel height as an etiology of Hallux Abductus 2 Valgus development: a electromagnetic static and 3 dynamic first metatarsophalangeal joint study”, In general, the paper is difficult of reading, it is necessary to increase the introduction and include more information about the background of the topic. The section of the method could be improved I observed some methodology errors and I don’t understand what is the aim of the study
- I can not find the section of sample size
- Part of the inclusion criteria are exclusion criteria
-The flow diagram is not correct, you have used a flow diagram of STROBE
- The result are difficult of understanding, with a lot of tables difficult to read.
You should be rewritten part of the paper to be accepted
Author Response
Thanks for your appraisals and comments in order to improve the quality of our manuscript. Deep and substantial modifications have been made according to your suggestions.
You can to check point by point all the corrections done it according with your comments and recomendations following the attached file, hoping it would be of your consent.

Reviewer 2 Report
The introduction does not cite or give enough credit to previous literature in this area ( i.e. other similar research work on relation between heel height and PPH and IMTT).
Section 2.2: Authors have to provide pictures of the device and experimental setup. It is difficult to follow as reader without a picturize of the experiments and sensor locations. Please also provide detailed specification about the sensors and validation. More specific information about the sensors and systems used for measurement in section 2.2 is needed ( both the 3 space and 6D electromagnetic systems needs to be clearly described).
What was the need to use 2 systems, please explain to the reader the clear rationale for using one for bone movement and one for Hallux? How validated are these systems?
Provide specification and details of the sensor and it accuracy, inherent error rate as per manufacturer (%) and if any validation was already carried out on the specific sensor used? Typically such sensors have upto 10% error from MEMS manufacturing process. Other error we introduce ( like mounting and positioning ) is on top of this inherent error. This is important because seems the result "PPH valgus and abduction deviation from 3.15±0.10º to 31 3.46±0.05º (p<0.05)" could become invalid this this error %margin was not taken into consideration.
Line 133 to 140 : Please provide picture of setup.
Line 148:152 : What was randomized? Heel height order? Was 5 trails done for each heel height? More clarity needed. IF order of heel height was not randomized, how was this effect to results handled in the statistics?
Procedures: How was the data from the measurement system post-processed? What filter was applied?
Does this local sample MDC take into account the inherent sensor error margins?
MDC is just a sample based statistical value to show difference. The authors also have to discuss in the significant differences "3.15±0.10º to 31 3.46±0.05º (p<0.05) and from 1.35±0.28º to 1.69±0.30º (p<0.001) respectively. In addition, a PPH 32 abduction increase from 1.01±0.36º to 1.31±0.46º (p < 0.05)" holds any clinical relevance (i.e. Minimal Clinically Important Difference (MCID)). Once MCID is determined, the authors also have to cite literature references to support their case.
Was the p-value corrected for multiple comparisons?
The authors state the following as the main purpose of the study "the purpose of this study was to know how the heel height may affect to PPH and 58 IMTT bones deviations, either in the three planes of motion in static condition and during the 59 dorsiflexion of the first MPJ sequence (heel-off phase of the dynamic condition), regardless of toe 60 box of the shoe, in order to see if there is some movement related to HAV deviation, which is 61 characterized by PPH in abduction (away to medial body line) in transverse plane and valgus 62 deviation in frontal plane toward the first MPJ (2), or / and adduction deviation of the IMTT (toward 63 medial body line) regarding to second MTT in transverse plane and presence of valgus in frontal 64 plane toward the first MCJ (2)."
The methods results and discussion should clearly address each aspect of this from the data obtained and outcome metrics calculated. In its present form of how the information is presented, the evidence for all these aims are not clerarly explained and communicated through the results and discussion. A rewriting of the results, discussion and the data tables presentation to make the reading easy is suggested.
Also, the authors have to justify using ICC for as a sole reliability metric. A high ICC does not always imply two variables are linearly related. A CCC (Concordance correlation) is better for reliability analysis.
Author Response
We appreciated all your contributions to improve our manuscript.
We hope that the deep modifications done will have the correct adjustment with your requirements.
Please, read the attachment file.
In addition, we incorporate language certificate..

Round 2
Reviewer 1 Report
Dear author, the suggestions that I recommended, you have done and these have improved the paper.
Author Response
Thank you very much for let us to improve the document and to get the correct level to publish it in the Journal.
Reviewer 2 Report
What is the actual sample size 80 or 96? Line 108 states 80 while the discussion first paragraph states n= 96 (line 398)?
Abstract: Methods: Please report sample size. "Crossover clinical trial". What was crossed over? The methods description in the main article does not seem to be reporting it as cross over clinical trial. Did the authors intend "Cross sectional"?.
The abstract terms "basal conditon" is it same as static condition (line 181). Please use consistent terms through out.
Once the above information is obtained, please check ( via statistics) if the results have effect of the randomized order?
Concordance correlation is different than Spearman's correlation. Please search literature for Lin's Concordance correlation (CCC). CCC is a better metric for such data rather than ICC or Spearman's.
Line 125:127: More clarity needed. Please rephrase (line 125:126), it is difficult to understand what is said. Also, please follow consistent units ( All mm, or cm or m). Switching within the same sentence makes it difficult to read.
Line 174:186: More clarity needed: How many steps from each trial was included for the analysis? Did a participant follow the same randomized order between the "with shoe" and unshod condition? or was there a within subject variation of the randomized order?
Was there randomized order followed for the static condition? If, so for each participant was it same as the dynamic or different from their dynamic randomized order?
Overall also address:
How was the order randomized, ( manual, program)? What were the orders chosen? Was the randomization done by someone blind to the outcome of the study? How many participants fell into each random order category? Highly recommend the authors the provide a flow chart of the experiment. To help authors with such a chart, please refer the STROBE statement for clinical trials.
Request authors to re-evaluate if there was an effect of the random order on the results.
Line 210: Please report the corrected p value. correct p=??.
Line 449: MDC only proves statistical validity, it need not necessarily have clinical relevant meaning. I suggest to include the term "...statistically valid " in line 449.
Line 396: Please rephrase "Job"
Line 452:453: Please rephrase for clarity.
Author Response
Thank You very much for your suggestions and corrections´ requirements. With them, you have let us to improve the level of the manuscript and to be possible it´s publication in this Journal.

Round 3
Reviewer 2 Report
Dear authors,
Thank you for making the suggested changes. I appreciate your time and effort for the same. I apologize that I missed to add an appreciation note in the previous round of review comments. The article looks good, however, there is some major concern of mine raised in previous review that has not been addressed sufficiently. Also here I provide some minor comments on consistency terminologies throughout the article. I will try my best to provide a more described concern on major concern and why it is vital to address it clearly.
Major comment:
Comment 1:
It is great that Figure 8 is added now on randomization. However my concern is the way the randomization is described is so unclear (Lines 170 : 185). To make my concern clear, as I understand from how it is described (in text and figure), there seems to be three layers of randomization, (layer 1: heel heights (line 170)) ; (Layer 2 : randomization between Static or Dynamic); (Layer 3: Randomization between four different walking conditions).
Also, line 178 : "this capture was repeated randomized for 5 trials....." . Not clear what the authors mean by this.
As described in the textual order,so the layer 1 is missing from fig 8?
Please declare based on you randomization how many of the n=80, did static as first choice or dynamic as first?
Now after this layer 2: there was randomization in 4 conditions (UNSHOD, Heel 3, Heel 6, Heel 9). What were the exact order (i.e. say Participant 1 : could have done Heel 9, heel 6, heel 3 and UNSHOD as their random order, then participant 2 some other order). How many such orders were there? 4 choice can lead to four factorial ( 4X3X2X1) number of order. So the authors should report how many orders were chosen and how many participants (n=??) went into each of these order.
Within the group who did dynamic as first, did participant do the dynamic and static trial in the same order, (i.e. for example if participant 1 performed Heel 9, heel 6, heel 3 and UNSHOD in Dynamic first, then when they did their static did participant 1 follow (Heel 9, heel 6, heel 3 and UNSHOD) or a different random order).
Since there is three layers of randomization, the authors have to clearly test the effect of order of experiment on the overall results. This requires a separate battery of statistical tests to study the effect of order based on how many random orders the authors chose to assign.
These descriptions of procedures are very important for someone to replicate these experiments.
If the authors feel there is a need to consult a statistician to describe this portion and test the effect of order on the results, I recommend that.
Comment 2:
Some of the correlations (for instance Table 2 PPH AD, PPH - ABD ( i.e. row 2 in table 2)), even though statistically significant. the correlation itself is very low.The statistical significancve could have been because of the sample size (n=80). Most correlation in (Table 2,3 6 and 7) are low to moderate even to statistically significant. The authors have to discuss about this in the discussion.
Other minor changes:
Some of the tables have "," instead of decimal points for numbers.
Please check the english thoroughly, many places it does not read right, for instance , line410.
I hope the authors are able to address these concerns.
Author Response
We want to thanks all the suggestions that you have done to our manuscript.
With the present corrections, especially done in material and methods section following your indications, we hope to reach the proper level to publish the paper on your journal.
We have attached the certificate of "American Manuscript Editors" in order to justify the english´s corrections, too.
Our kindly regards.

Round 4
Reviewer 2 Report
Report 1:
Dear authors,
Thank you for making the modifications and providing clarifications to concerns. I appreciate your time and effort in doing so. However, the updated manuscript does not contain the modifications as stated in the 'response to reviewers' document.
Some additional comments based on the pdf version uploaded:
Although the response for comment (2) states that line 173:178 has nerw text, the pdf downloaded does not reflect those changes made.
Changes to limitation section as mention in the 'response to reviewers: item 6' is also not seen in the uploaded manuscript sent for review. Line 460 -471 is already refernce list in the uploaded version. Please check the uploaded version is the right version.
within this please add references to " Most of the main limitations in other similar studies......."
The manuscript still has formatting consistency and can benefit in terms of clarity from english fixes. For example
They had similar Spearman values in the present work but did show any contrast between data because they did not assess individual movements of any segment bone as PPH or I MTT.
the formatting for metrics are different, for instance 6cm, 6-cm, 6 cm , six cm. Please be consistent with all such metrics throughout the manuscript.
It is not clear if the ICC is the tables are the Spearman's correlation. Please clarify this in text
(line 196) and table abbreviations. Also, please rephrase line 218: Did the authors mean to say both CCC and Spearman's were carried out?
CCC to be added to table abbreviations.
Please add literature reference to CCC.
Please rephrase line 218: Did the authors mean to say both CCC and Spearman's were carried out?
Please mention n=40 in figure 8.
Effect of order is independent of balanced sample size. This is a weak hypothesis to make,
Please acknowledge this in the limitation: We were not able to study the effect of order on our sample because
"didn´t write the different orders of each one neither the number of these selections. We assumed the possible "order effect" as "perfectly balanced" because all the study subgroups of each station have the same number of subjects and this can dilutes the “order effect”. Future study design should include a section of studying the effect of order of experiments.
--------------------------------------------------
Report 2
I have some comments in the pdf document.
In addition, the comments I already provided is still valid and has to be addressed.
Author Response
We attach the response document in that file.
In addition, we attach another two documents:
- a word marked copy, with yellow marks on the text in order to highlight the corrections made
.- a pdf clean version
We want to grateful your comments to improve the level of the manuscript, hoping the acceptance to the publication. Thank you very much.
